# Experimental and Theoretical Studies of Sonically Prepared Cu–Y, Cu–USY and Cu–ZSM-5 Catalysts for SCR deNOx

**Przemysław J. Jodłowski** [1,*,†] **, Izabela Czekaj** [1,*,†] **, Patrycja Stachurska** [1] **, Łukasz Kuterasiński** [2] **, Lucjan Chmielarz** [3] **, Roman J. Jędrzejczyk** [4] **, Piotr Jeleń** [5] **, Maciej Sitarz** [5] **, Sylwia Górecka** [6] **, Michal Mazur** [7] **and Izabela Kurzydym** [1]

1   Faculty of Chemical Engineering and Technology, Cracow University of Technology, Warszawska 24, 31-155 Kraków, Poland; patrycja.stachurska@gmail.com (P.S.); izabela.kurzydym@doktorant.pk.edu.pl (I.K.)
2   Jerzy Haber Institute of Catalysis and Surface Chemistry, Polish Academy of Sciences, Niezapominajek 8, 30-239 Kraków, Poland; nckutera@cyf-kr.edu.pl
3   Faculty of Chemistry, Jagiellonian University, Gronostajowa 2, 30-387 Kraków, Poland; lucjan.chmielarz@uj.edu.pl
4   Małopolska Centre of Biotechnology, Jagiellonian University, Gronostajowa 7A, 30-387 Kraków, Poland; roman.jedrzejczyk@uj.edu.pl
5   Faculty of Materials Science and Ceramics, AGH University of Science and Technology, al. Mickiewicza 30, 30-059 Kraków, Poland; pjelen@agh.edu.pl (P.J.); msitarz@agh.edu.pl (M.S.)
6   Institute of Environmental Technology, CEET, VSB—Technical University of Ostrava, 17. Listopadu 15/2172, 708 00 Ostrava, Czech Republic; sylwia.gorecka@vsb.cz
7   Department of Physical and Macromolecular Chemistry, Faculty of Science, Charles University, Hlavova 8, 128 43 Prague, Czech Republic; michal.mazur@natur.cuni.cz
*   Correspondence: przemyslaw.jodlowski@pk.edu.pl (P.J.J.); izabela.czekaj@pk.edu.pl (I.C.); Tel.: +48-12-6282760 (P.J.J.); +48-12-6282111 (I.C.)
†   P.J. Jodłowski and I. Czekaj declare to be equally first authors.

**Abstract:** The objective of our study was to prepare Y-, USY- and ZSM-5-based catalysts by hydrothermal synthesis, followed by copper active-phase deposition by either conventional ion-exchange or ultrasonic irradiation. The resulting materials were characterized by XRD, BET, SEM, TEM, Raman, UV-Vis, monitoring ammonia and nitrogen oxide sorption by FT-IR and Diffuse Reflectance Infrared Fourier Transform Spectroscopy (DRIFTS). XRD data confirmed the purity and structure of the Y/USY or ZSM-5 zeolites. The nitrogen and ammonia sorption results indicated that the materials were highly porous and acidic. The metallic active phase was found in the form of cations in ion-exchanged zeolites and in the form of nanoparticle metal oxides in sonochemically prepared catalysts. The latter showed full activity and high stability in the SCR deNOx reaction. The faujasite-based catalysts were fully active at 200–400 °C, whereas the ZSM-5-based catalysts reached 100% activity at 400–500 °C. Our in situ DRIFTS experiments revealed that Cu–O(NO) and Cu–NH₃ were intermediates, also indicating the role of Brønsted sites in the formation of NH₄NO₃. Furthermore, the results from our experimental in situ spectroscopic studies were compared with DFT models. Overall, our findings suggest two possible mechanisms for the deNOx reaction, depending on the method of catalyst preparation (i.e., conventional ion-exchange vs. ultrasonic irradiation).

**Keywords:** zeolites; deNOx; sonication; DFT; reaction mechanism; copper catalysts





## 1. Introduction

One of the problems that are directly associated with environmental protection are air pollutants [1]. For example, nitrogen oxides (NOx) harm human health and life in general, forming photochemical smog due to solar radiation. In addition, nitrogen oxides emitted in effluent gases may derive from various sources, such as the automotive, energetics and heavy industry [2]. However, their levels can be lowered by selective catalytic reduction (SCR; deNOx) [3]. In SCR, Cu-modified zeolites stand out as effective deNOx catalysts with

various reducing agents: ammonia [4–10], methane [11], propane [11–14], propene [15], and other hydrocarbons [16]. In addition, hydrocarbons can be used as reducing agents in deNO$_x$ to simultaneously remove NO$_x$ and Volatile Organic Compounds (VOCs).

When developing highly active and selective zeolite catalysts, several factors, such as zeolite type, structure and acidity/basicity, must be considered [17]. Over the years, zeolites have been used as an alternative to commercial V$_2$O$_5$/WO$_3$–TiO$_2$ catalysts. Iwamoto et al. [18] studied the activity of C$_2$H$_4$ as a reducing agent and observed the following activity sequence: Cu–ZSM-5 > Cu–MOR = Cu–FER > Cu–Y. Subsequently, Halasz et al. [19] suggested that the bridging hydroxyl groups present in the ZSM-5 structure are active sites of NO oxidation by O$_2$ to surface nitrates. B. Pereda-Ayo et al. [8] and Putluru et al. [9] studied deNO$_x$ by a Cu-promoted zeolite SCR catalyst in NH$_3$, showing that the content of the active phase varied with the structure of the zeolite in ion-exchanged samples. The highest and the lowest amounts of Cu were found in the Y and ZSM-5 zeolites, respectively. Moreover, these results matched the Si/Al ratio. The higher the zeolite Si/Al ratio was, the lower the content of the active phase (metal species) introduced into the zeolitic structure would be, thus showing an inverse correlation between the zeolite Si/Al ratio and active phase content. The interesting review on decomposition of nitric oxide on Cu–Y and Cu–ZSM-5 is presented by Davydov [20]. The different pathways on NO decomposition are suggested; however, in the case of the deNO$_x$ process, still it is not evident the role of the zeolite preparation method in differences in NO decomposition and the deNO$_x$ reaction mechanism.

The zeolite preparation method also affects catalyst performance. De la Torre et al. [21] prepared Cu–ZSM-5 and Cu–BEA zeolites, either by ion-exchange or by impregnation, with copper content ranging from 1 to 6%. At a higher metal content, the maximum NO$_x$ conversion decreased, although this was achieved at a lower temperature, most likely due to NO oxidation to NO$_2$ and subsequent activation of a fast SCR reaction. In turn, B. Pereda-Ayo et al. [8] assessed the effect of the monolith preparation sequence on catalyst performance. As a general trend, monoliths prepared by zeolite ion-exchange followed by monolith washcoating reached both higher NO$_x$ conversion and selectivity to N$_2$ than catalysts prepared in the reverse order, thus matching the higher amount of copper and a better distribution of the metal over the washcoat. Most importantly, Cu-modified zeolites prepared by wet impregnation or ion exchange show high activity in deNO$_x$ [12,22–25], but in contrast to the previous methods, these techniques are widely used to prepare zeolitic catalysts for deNO$_x$ processes.

Recently, sonochemical methods were efficiently used to prepare active nanocrystalline catalysts for gas exhaust abatement [26]. Thus, ultrasound-assisted synthesis could be an efficient method for the production of zeolite-based catalysts for deNO$_x$ reactions. Ultrasonic treatment accelerates active-phase dispersion on the surface of the carrier, facilitating the attachment of nanoscale components to its structure, thereby improving the catalytic properties of the resulting catalyst. In addition, the use of ultrasound techniques during the impregnation process significantly accelerates catalyst modification [27]. Several researchers have investigated ultrasound effects on different chemical processes, including numerous syntheses of various amorphous and crystalline chemicals, assessing significant changes in both ultrasound-assisted processes and properties of reaction products. For instance, sonochemistry can be used for polymer preparation and modification, metal oxide nanoparticle synthesis and the homogenization of liquids, among many other applications [28].

Several reports on the application of ultrasound irradiation to zeolite preparation are available in the literature. Andac et al. [29] performed an ultrasonic-assisted synthesis of A-type zeolite in sodium aluminosilicate solution, resulting in a product with a highly crystalline phase and with the desired zeolite structure. In another study, Andac et al. [29] investigated the direct synthesis of zeolite 4A coatings on stainless steel carriers, finding that more nuclei formed more quickly after ultrasound treatment based on structural and morphological analyses. A similar effect was observed during the preparation of MCM-22 in which the ultrasonic-assisted aging of the initial aluminosilicate gel shortened

the crystallization time of MCM-22 [30]. Belviso et al. [31] performed ultrasound-assisted synthesis of zeolites from fly ash at a low temperature range (25–60 °C), also noting that ultrasonic treatment accelerated zeolite formation at a low temperature (25 °C), as shown by XRD and SEM. In another study, these authors [32] synthesized zeolites under hydrothermal conditions with seawater from fly ash at low temperature, observing that the ultrasound treatment decreased the crystallization temperature, albeit worsening the crystallinity of a final product when replacing distilled water with seawater. Pal et al. [33] prepared NaP zeolite nanocrystals using ultrasound for 3 h at room temperature (no autogenous pressure or hydrothermal treatment), assessing the effect of both ultrasonic energy and irradiation time on the crystallinity and purity of the final product, that is, increasing the energy of the ultrasound waves caused worsened crystallinity without changing the phase purity, whereas simultaneously increasing the irradiation energy and time slightly increased the crystallinity of the final product.

Based on the above, the objective of this study was to prepare and characterize the Cu/ZSM-5, Cu/Y and Cu/USY zeolite catalysts for the selective catalytic reduction of NO by sonochemical irradiation. Furthermore, sonicated catalysts were compared with conventional ion-exchange catalysts. All samples were characterized using various methods, such as atomic absorption spectrometry (AAS), X-ray diffraction (XRD) and low temperature sorption of liquid nitrogen. Moreover, the sorption and acidic properties were determined by in situ InfraRed (IR) spectroscopy, using $NH_3$, CO and NO as probe molecules, studying catalytic activity in the SCR of NO with $NH_3$ as a reducing agent. The results from the experimental in situ spectroscopic studies were supported by DFT models to compare at the molecular level differences in reaction mechanisms on (i) differently prepared catalysts (i.e., conventional ion-exchange vs. ultrasonic irradiation), and (ii) differences between Cu/Y and Cu/ZSM-5.

## 2. Results

### 2.1. Catalysts Preparation and Characterization

The parent zeolite samples were prepared according to the method described in detail in our previous paper [34] and in Section 4. Materials and Results. Several experimental techniques were used for catalyst characterization: XRD, SEM, HRTEM, UV-Vis DR, FT-IR, and in situ DRIFT. The NO and $NH_3$ evolution on the dehydrated catalysts at different temperatures as well as SCR deNO$_x$ were performed. Details on catalyst preparation and characterization are presented in Table 1.

**Table 1.** Catalysts and their physicochemical properties.

| Catalyst | Preparation Method | Si/Al | Copper Content, wt% | $S_{BET}$, m²/g | $V_P$ Total, cm³/g | Acid Sites Conc. ($NH_3$ sorption), µmol/g$_{cat.}$ | | Active Cu Lewis Sites Conc. (CO Sorption), µmol/g$_{cat.}$ |
|---|---|---|---|---|---|---|---|---|
| | | | | | | **Brønsted** | **Lewis** | |
| Cu/Y | Ion exchange | 4.52 | 6.50 | 281.0 | 0.21 | 49 | 2116 | 72 |
| Cu/Y/s | Sonication | 4.52 | 3.82 | 137.4 | 0.12 | 70 | 2153 | 64 |
| Cu/USY | Ion exchange | 4.52 | 4.91 | 439.6 | 0.34 | 21 | 1452 | 184 |
| Cu/USY/s | Sonication | 4.52 | 3.72 | 450.8 | 0.34 | 63 | 1518 | 90 |
| Cu/ZSM-5-15 | Ion exchange | 15 | 0.62 | 283.1 | 0.26 | 229 | 355 | 196 |
| Cu/ZSM-5-15/s | Sonication | 15 | 0.10 | 317.0 | 0.28 | 272 | 365 | 60 |
| Cu/ZSM-5-37 | Ion exchange | 37 | 0.32 | 368.5 | 0.37 | 173 | 225 | 49 |
| Cu/ZSM-5-37/s | Sonication | 37 | 0.13 | 359.1 | 0.35 | 220 | 148 | 40 |

To determine the impact of ultrasound on the structure of the materials, HRTEM analyses were performed, and the results are presented in Figure 1.

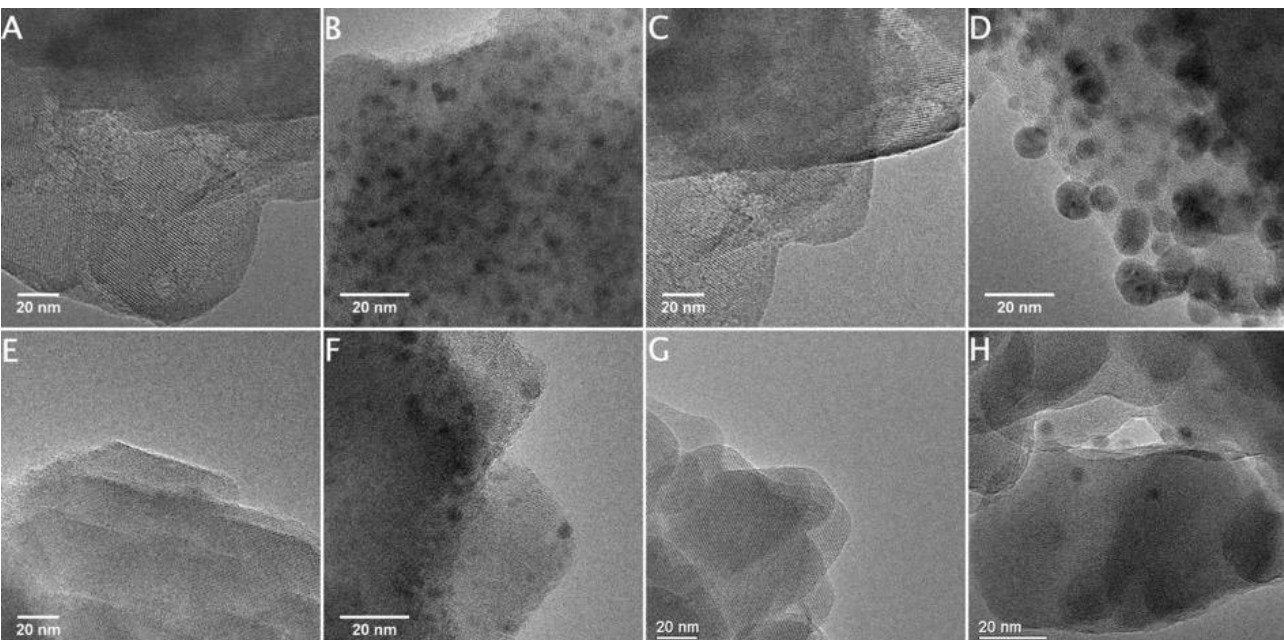

**Figure 1.** HRTEM images of the catalysts: (**A,B**) CuYs, (**C,D**) Cu/USYs, (**E,F**) Cu/ZSM-5-15s. (**E–H**) Cu/ZSM-5-37s.

The catalysts were also examined by µRaman spectroscopy (Figure 2A,B). The µRaman spectra of Y/USY based zeolites revealed bands at 220, 322, 384, 495 and 660 cm$^{-1}$, albeit only for Cu/USY/s (Figure 2A). The analysis of the Raman spectra of Cu modified ZSM-5 zeolites reveals four bands at 290, 379, 468 and 800 cm$^{-1}$ (Figure 2B). These bands may be due to framework vibrations.

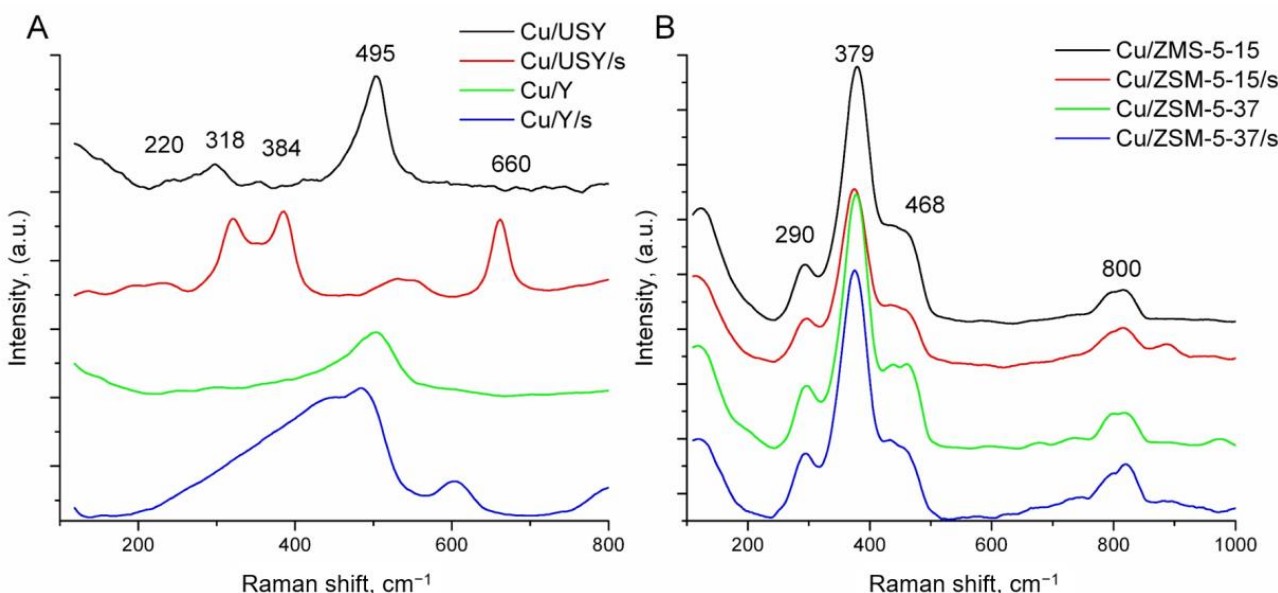

**Figure 2.** µRaman spectra of the samples; 488 nm line; (**A**) Cu/Y and Cu/USY, (**B**) Cu/ZSM-5-15 and Cu/ZSM-5-37.

The results from the static $NH_3$ sorption experiments are presented in Figure 3A,B.

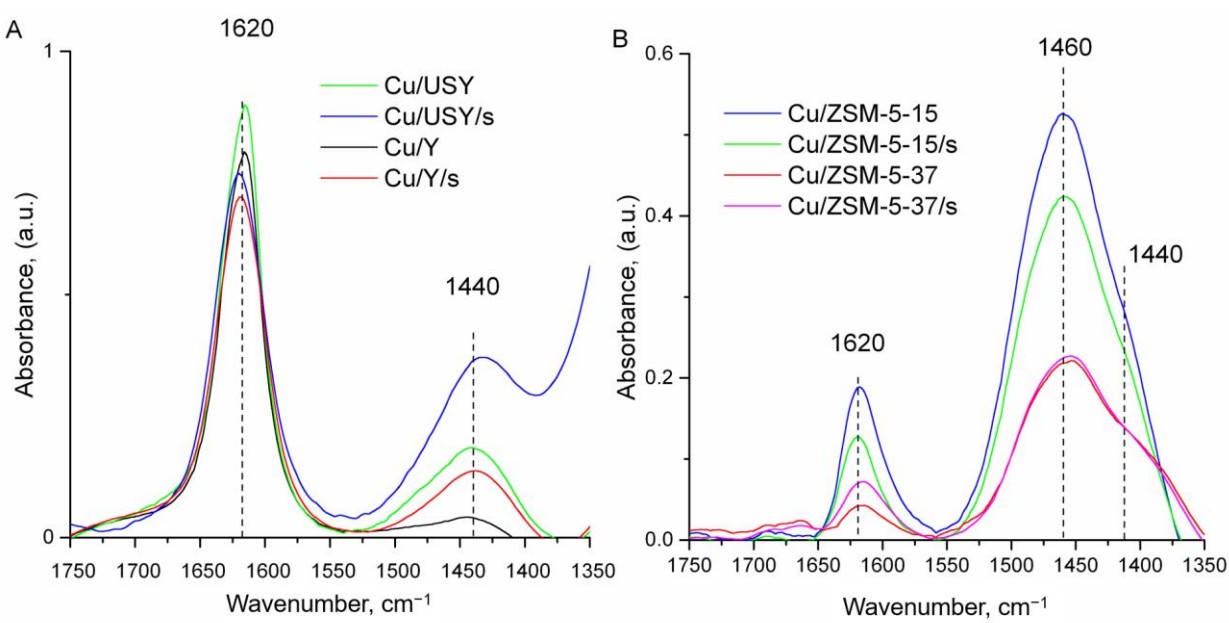

**Figure 3.** In situ FTIR spectra of zeolite samples with adsorbed NH₃ at room temperature: (**A**) Cu/Y and Cu/USY, (**B**) Cu/ZSM-5-15 Cu/ZSM-5-37.

The results of NO adsorption are presented in Figure 4.

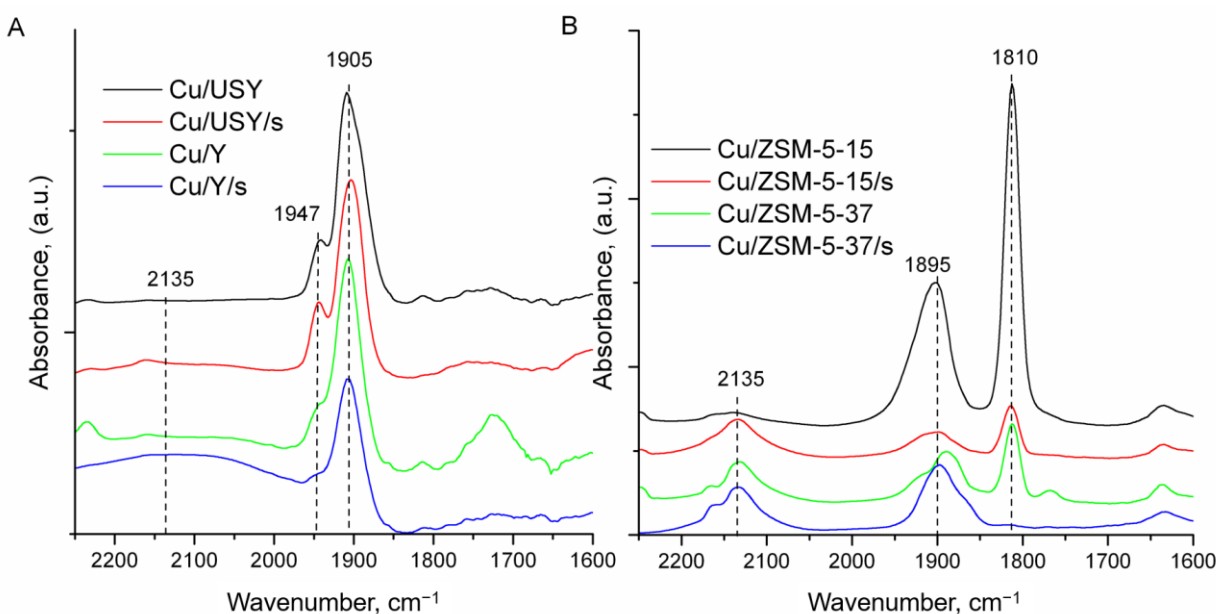

**Figure 4.** In situ FTIR spectra of the zeolite samples with adsorbed NO at room temperature: (**A**) Cu/Y and Cu/USY, (**B**) Cu/ZSM-5-15 Cu/ZSM-5-37.

### 2.2. Activity Measurements

The results of catalytic activity in SCR deNO$_x$ are expressed as conversion/selectivity profiles in Figure S9 and as specific activity—turnover frequency (TOF)—in Figure 5. Complete NO conversion was achieved with all catalyst samples (Figure S9). However, for USY- and Y-based catalysts, the temperature region in which the catalysts worked effectively was broad, ranging from 200 to 400 °C, whereas their ZSM-5-based counterparts showed full conversion in a narrow temperature range shifted to higher values, i.e., 375–500 °C.

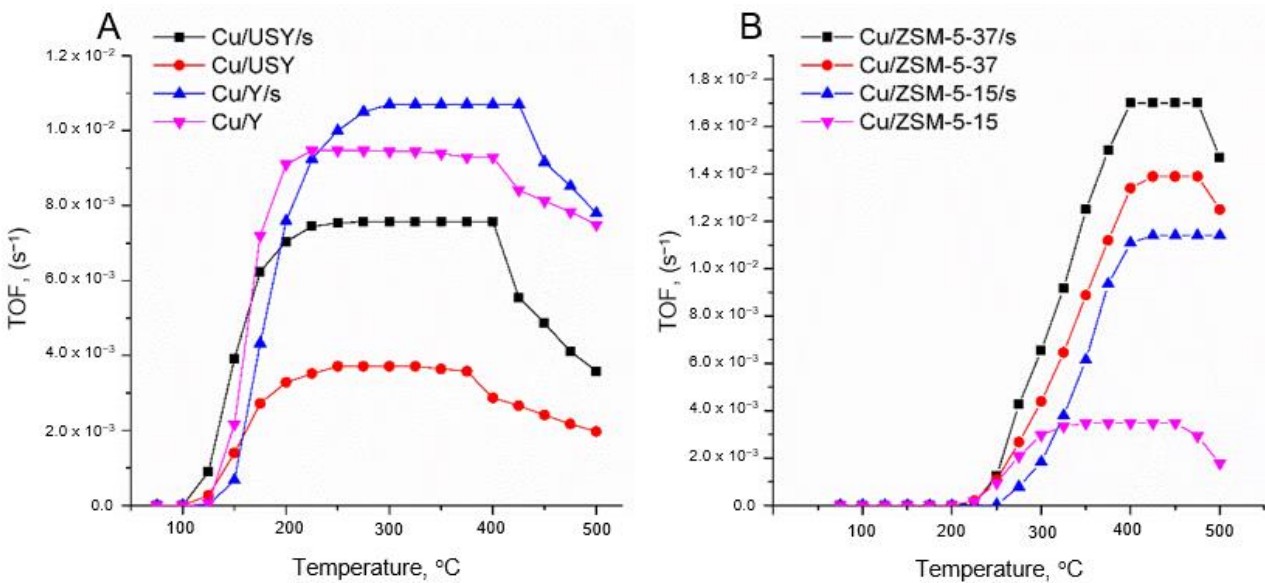

**Figure 5.** Reaction rate expressed as TOF: (**A**) Cu/Y and Cu/USY samples, (**B**) Cu/ZSM-5-15 and Cu/ZSM-5-37 samples.

### 2.3. In Situ DRIFT Studies

Dynamic in situ DRIFT studies were performed to determine the surface species during the SCR deNO$_x$ reaction. The experiments included the temperature programmed surface reaction (TPSR) of NO$_x$ and NH$_3$ and TPSR SCR deNO$_x$ reaction, and the results are presented in Figures S10 and S11 for Y/USY and ZSM-5 catalysts, respectively. Considering the above, we present the zoomed areas for the OH region and the Brønsted and Lewis regions for ammonia adsorption (Figure 6).

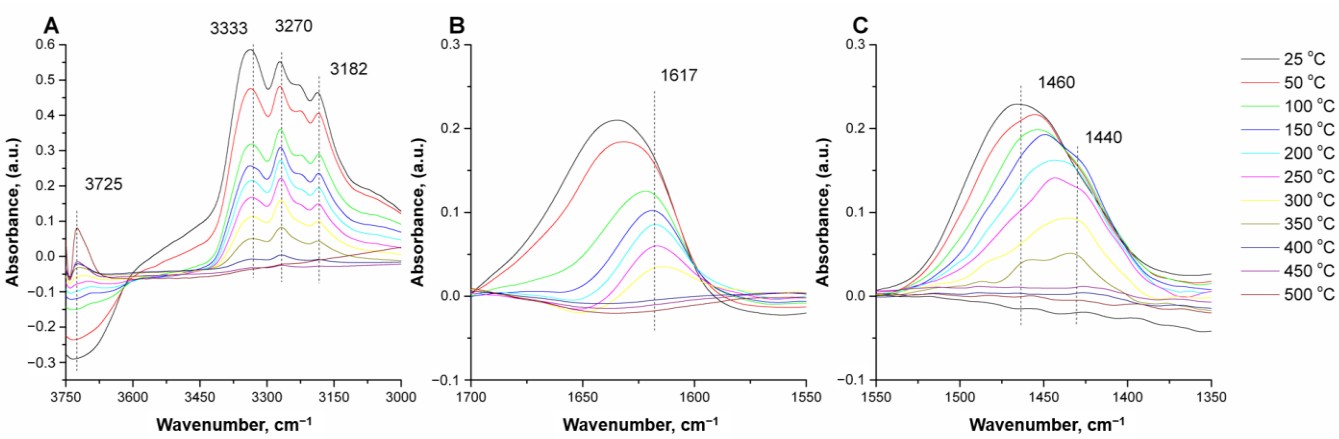

**Figure 6.** DRIFT spectra for Cu/Y/s catalysts during NH$_3$ dynamic sorption experiments in different wavenumber regions: (**A**) 3000–3750 cm$^{-1}$, (**B**) 1550–1700 cm$^{-1}$ and (**C**) 1350–1550 cm$^{-1}$.

### 2.4. Theoretical Modeling

Cluster models of Y/USY (Al$_2$Si$_{22}$O$_{66}$H$_{36}$, which includes 24T positions) and ZSM-5 (Al$_2$Si$_7$O$_{25}$H$_{15}$, which includes 9T positions) zeolite structures were used with Cu particles adsorbed on aluminum centers in the faujasite framework (Figure S1).

The oxidized and hydrated Cu monomers and Cu$_2$O dimers (Figure 7) were analyzed, according to previous studies of iron complexes [35]. The cluster models are as large or larger than those reported for NO adsorption studies in the literature (9T positions) [36,37].

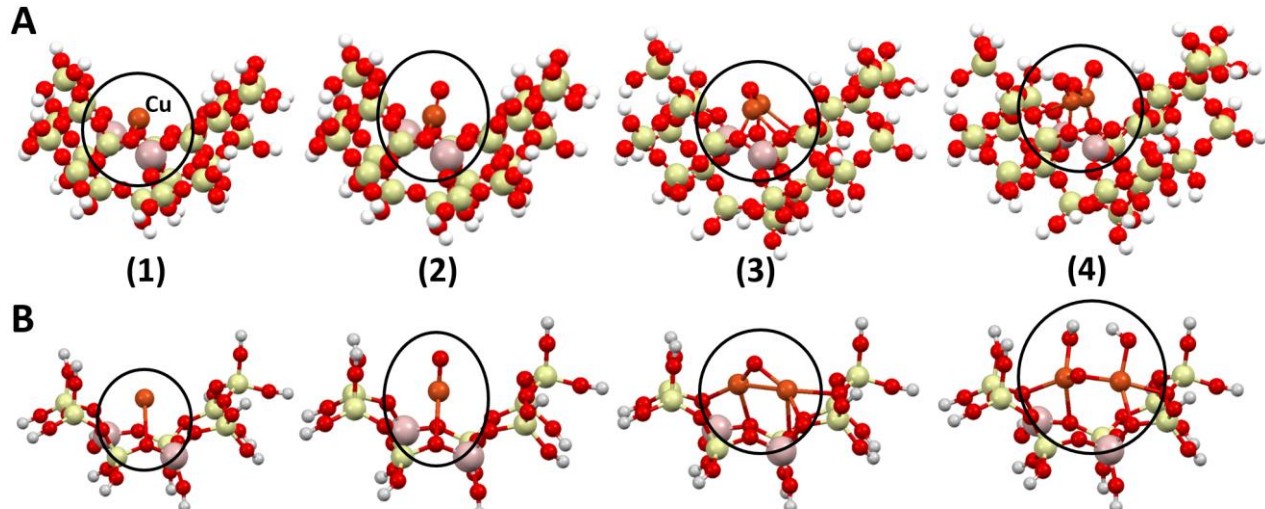

**Figure 7.** Copper models in (**A**) Y/USY and (**B**) ZSM-5. (1) Cu, (2) CuO monomer, (3) $Cu_2O$ dimer and (4) $Cu_2O(OH)_2$ hydrated dimer; Cu—orange, Si—yellow, Al—pink, O—red, H—white.

## 3. Discussion

### 3.1. Catalyst Characterization

The results from the XRD analyses are presented in Figure S2. The diffractograms of the zeolites were compared with those available in the International Zeolite Database [38], confirming the presence of the specific zeolite phase. As shown in Figure S2, the crystalline structure of catalysts prepared by sonochemical route remains unchanged. Furthermore, no differences in XRD patterns are noted between zeolite structures from the IZA database and those of the catalysts prepared in this study, which may suggest the atomic dispersion of copper in both preparation methods.

The results from the catalyst characterization analyses by XRF and BET are summarized in Table 1. As expected, small differences in catalyst loading are identified between ion-exchanged and sonically prepared samples. In all zeolite catalysts, copper loading is higher in ion-exchanged samples. The copper content was 6.50 and 3.82 wt% for Cu/Y and Cu/Y/s catalysts, and 4.91 and 3.72 wt% for Cu/USY and its sonochemically prepared counterparts, respectively. Differences in catalyst loading between the ion-exchanged catalysts and their sonochemically prepared counterparts can be due to the differences in the ion-exchange/sonication time, or/and in the interactions of the catalyst precursors during the impregnation/sonication procedure. In the ion-exchange method, the counterion atom present in the zeolite crystal can be exchanged by the metal ions present in the catalyst precursor (commonly a metal nitrate). This metal ion compensates for the negative charge of the zeolite framework. Conversely, in the sonochemical preparation method, the interaction between the catalyst precursor and the ultrasound forms a suspension of reduced/oxidized nanoparticles. Some metal ions may be present in the metal precursor, which may further interact with the cation exchange positions available in the zeolite structure, but the reduced/oxidized nanoparticles are in excess.

In the ZSM-5 zeolite catalyst, the copper content is considerably lower than in Y and USY zeolites. Copper loading ranges from 0.62 to 0.10 wt% for Cu/ZSM-5-15 and Cu/ZSM-5-15/s samples, respectively. Additionally, for Cu/ZSM-5-37 and Cu/ZSM-5-37/s, this value is 0.32 and 0.13 wt%, respectively. The considerable difference in catalyst loading between Y/USY and ZSM-5 zeolites results from the lower alumina content of the zeolite framework [39]. The Si/Al ratios increase in the following order: Y = USY < ZSM-5-15 < ZSM-5-37 (cf. Table 1).

The catalyst porosity was determined by nitrogen adsorption. The $S_{BET}$ surface area and total pore volume are summarized in Table S1. For the catalyst pairs Cu/Y and Cu/Y/s, $S_{BET}$ was 281.0 and 137.4 $m^2/g$, respectively. Simultaneously, ultra-stabilization noticeably increased the surface area of the zeolite. The BET surface area of Cu/USY and

Cu/USY/s ranged from 439.6 to 450.8 m$^2$/g, respectively. For the ZSM-5 catalysts, no significant difference was found between the samples. The $S_{BET}$ of Cu/ZSM-5-15 and Cu/ZSM-5-15/s ranged from 283.1 and 317.0 m$^2$/g, whereas the specific surface area of the ZSM-5 zeolites with a Si/Al = 37 was slightly higher, at 368.5 and 359.1 m$^2$/g for Cu/ZSM-5-37 and Cu/ZSM-5-37/s, respectively.

The SEM/EDX results are presented in both SEM micrographs (Figures S3 and S4) and SEM/EDX maps (Figures S5 and S6). The SEM images show no significant differences between crystal shapes. In all SEM micrographs (Figures S3 and S4), no crystal shape can be specifically assigned, but the structure of the ZSM-5 zeolites is denser (Figure S4). The comparison between SEM images of Y/USY and ZSM-5 shows that the Y/USY zeolite crystals are smaller and have a more regular shape. The distribution of copper in the catalysts prepared by ion exchange and sonochemical irradiation is presented in Figures S5 and S6, respectively, showing that the zeolites are uniformly covered with copper in oxidized form.

Detailed HRTEM analysis shows that sonochemical preparation of catalysts leads to the formation of well-defined spherical copper nanoparticles (visible as dark spots) uniformly distributed over the zeolite surface. A considerable higher amount of copper nanoparticles is observed in faujasite- than in ZSM-5-based catalysts, in line with the quantitative AAS results summarized in Table S1. The average size of the copper nanoparticles in Y and ZSM-5-based catalysts is visibly smaller than that of USY.

To determine the state of the copper oxide introduced into the zeolite, either by ion exchange or by sonication, in situ UV-Vis and in situ μRaman analyses were performed, and their results are presented in Figures S7 and 2, respectively. The UV-Vis spectra of the copper zeolites show an intense broad band at 200–400 nm and a weak broad band at 600–850 nm. The deconvoluted Cu/Y UV-Vis spectra (Figure 6 inset) consist of three major band regions: <250 nm, the band centered at 270 nm, and the high wavelength region of >330 nm. The low wavelength band below 250 nm can be attributed to Cu$^+$ species distributed over the surface of the catalyst [40], whereas the band centered at 270 nm can be assigned to the charge transfer between monomeric Cu$^{2+}$ and oxygen atoms [41]. The weak band at 330 nm is due to [Cu–O–Cu]n-type clusters on the zeolite surface [42]. The weak broad band at 600–850 nm derives from the d–d transition of octahedrally coordinated CuO. The band located at approximately 700 nm is related to the d–d transition of Cu$^{2+}$ ions in pseudo-octahedral coordination (e.g., Cu(H$_2$O)$_6$$^{2+}$) [43–45].

The bands at μRaman spectra (Figure 2A,B) at 318 and 384 cm$^{-1}$ are associated with A$_g$ and B$_g$ modes, characteristic for CuO [46]. The intense band at 495 cm$^{-1}$ may derive from the 4-membered ring bending [47]. The additional bands appearing at the 300–600 cm$^{-1}$ region are the structure-sensitive Raman bands that can be assigned to in-plane motion on the oxygen atoms perpendicular to the T–O–T (T = Si, Al) bond [47]. The band at 660 cm$^{-1}$ may be attributed to B$_g$ mode of CuO [46]. The bands at 290, 379 and 468 cm$^{-1}$ at Figure 2B belong to the bending modes of 6-, 5- and 4-membered rings of zeolite frameworks, whereas the band at 800 cm$^{-1}$ is assigned to T–O–T symmetrical stretching vibrations [47].

To find the correlation between the structure of the samples and their catalytic activity, in situ FT-IR analyses of probe molecules adsorbed over the surface of the samples were performed, and the results are presented in Table 1. The concentration of acid sites was determined for both Y and USY catalysts by NH$_3$ adsorption, showing that Y-based catalysts have a considerably lower concentration of Brønsted acid sites than their ZSM-5 analogs. The high copper loading of the Y and USY samples may decrease the concentration of Brønsted acid sites. In turn, their concentrations of Lewis acid sites are one order of magnitude higher than those of ZSM-5 zeolites, due to differences in the Si/Al ratio between all samples and the high copper loading of Y and USY zeolites (Table 1). Y and USY zeolites have low Si/Al values, thus matching the high number of possible cation exchange sites. Conversely, the high Si/Al values of ZSM-5-based zeolites (Si/Al = 15 and 37) result in a noticeable decrease in possible cation exchange sites. The use of probe molecules in FT-IR also provides the necessary information about the nature of the copper active sites introduced into the zeolite. By using ammonia as the probe molecule, all types of Lewis and

Brønsted acid sites can be recognized, thereby differentiating the type of copper centers: $Cu^{2+}$ and $Cu^+$ present in exchange positions and Cu dispersed over zeolite surface, which react with ammonia and function as Lewis acid sites. Additionally, ammonia may also react with the residual hydroxyl groups, forming $NH_4^+$ cations [48].

The adsorption spectra of ammonia on Cu/Y and USY zeolites prepared by both ion exchange and sonochemical irradiation (Figure 3) reveal two bands at 1620 and 1440 cm$^{-1}$. The band at 1440 cm$^{-1}$ can be attributed to $NH_4^+$ species, formed after the reaction of $NH_3$ with Brønsted acid sites, whereas the band at 1620 cm$^{-1}$ confirms the presence of Lewis acid sites originating from copper ions introduced into the zeolite [49]. The typically narrow band at 1620 cm$^{-1}$ may suggest the presence of homogeneous Lewis acid sites with only one type of copper center [50]. Conversely, for Cu–ZSM-5 zeolites, the broad band at ~1460 cm$^{-1}$ splits into two bands at 1460 and 1440 cm$^{-1}$. The band splitting at 1440 cm$^{-1}$ may occur in zeolites with two types of Brønsted acid sites [51]. The comparison between $NH_3$ spectra obtained for Cu-modified Y/USY and ZSM-5 zeolites (Figure 4A,B) shows that the number of Lewis acid sites (band 1620 cm$^{-1}$) is considerably higher in Y and USY catalysts than in their ZSM-5 based counterparts and that the concentration of Lewis acid sites decreases in the following order: Cu/ZSM-5-15 > Cu/ZSM-5-15/s > Cu/ZSM-5-37 > Cu/ZSM-5-37/s. In turn, the reverse trend is observed when comparing the abundance of Brønsted acid sites (broad band at ~1440 cm$^{-1}$). Since ammonia covers all Lewis acid centers, without differentiating copper from alumina species, the sorption of CO and NO probe molecules was analyzed because CO is known to selectively adsorb to $Cu^+$ centers, whereas NO covers both $Cu^+$ and $Cu^{2+}$ sites [52]. The CO and NO sorption results are presented in Figure S8 and Figure 5, respectively, the former showing two bands at 2155 and 2140 cm$^{-1}$ (originating from CO vibrations) for Cu/USY and Y zeolites (Figure S8A) assigned to non-equivalent $Cu^+$ sites as previously reported by Datka and Kozyra [53], who identified two types of $Cu^+$ sites in the CuY zeolite. According to these authors, the low-frequency bands at 2140 cm$^{-1}$ strengthened the π back and weakened π donation, with the reverse trend for the high-frequency band. The analysis of IR spectra of ZSM-5-based catalysts (Figure S8B) shows that the CO molecule is bound at one site, and only one band at 2155 cm$^{-1}$ can be detected. To characterize the $Cu^{2+}$ sites present in the samples, NO sorption was analyzed but the FT-IR spectra were more complex because NO adsorbs to both $Cu^+$ and $Cu^{2+}$.

The broad band appearing at 2135 cm$^{-1}$ is detected in all samples (Figure 4). This band is characteristic of $NO^+$ adsorbed to the surface of the catalysts [54–56]. When analyzing the NO adsorption spectra of USY and Y-based zeolites, two main bands at 1947 and 1905 cm$^{-1}$ were detected. These bands can be attributed to $Cu^{2+}$–NO adducts [57]. Small bands appearing in the range of 1700–1850 cm$^{-1}$ can be assigned to both $Cu^+$–NO and $Cu^{2+}$–$(NO)_2$ species. In the analysis of NO adsorption spectra for ZSM-5-based catalysts, the bands of the major nitrosyl complexes $Cu^{2+}$–NO and $Cu^+$–NO are shifted to lower wavenumbers (Figure 4B). The main bands appear at 1895 and 1810 cm$^{-1}$ and may come from $Cu^{2+}$–NO and $Cu^+$–NO, respectively [58]. The intense and sharp band at 1810 cm$^{-1}$ may indicate that $Cu^+$ predominantly occurs on the surface of the catalysts. This phenomenon changes with the decrease in the concentration of Lewis acid sites (cf. Table 1). Quantitative Lewis acidity for ZSM-5-based catalysts changes in the following order: Cu/ZSM-5-15 > Cu/ZSM-5-15/s > Cu/ZSM-5-37 > Cu/ZSM-5-37/s. For Cu/ZSM-5-37/s, the band at 1810 cm$^{-1}$ was not detected. This effect may be explained by rapid $Cu^+$ oxidation to $Cu^{2+}$ by NO.

### 3.2. Activity Measurements

Among the USY and Y catalysts, the Cu/USY/s catalyst showed the highest activity, with maximal conversion at 225 °C, at similar levels to its counterpart prepared by conventional ion exchange—Cu/USY and Cu/Y catalysts. In turn, the Cu/Y/s catalyst prepared by sonication had the lowest activity. Moreover, the most active catalyst (Cu/USY/s) had

the lowest copper content (cf. Table 1), but comprehensively determining catalytic activity in the SCR deNO$_x$ reaction requires considering the selectivity to N$_2$.

The analysis of selectivity profiles of the set of USY and Y catalysts showed that the catalysts prepared by sonication had the highest selectivity. Their selectivity to N$_2$ along the reaction remained almost unchanged, with only a small decrease to 97% at the highest reaction temperature (Figure S9A). The selectivity of USY and Y catalysts prepared by conventional ion exchange considerably decreased to 94 and 95% at 400 and 450 °C, respectively. From the group of ZSM-5-based catalysts, Cu/ZSM-5-15 had the highest activity, whereas the other catalysts of this series showed similar activity. However, when considering selectivity to N$_2$ during this process, Cu/ZSM-5-15 had the lowest selectivity, whereas the other catalysts maintained 98–100% selectivity, albeit slightly decreasing at 500 °C. A more in-depth analysis of the conversion profiles over ZSM-5 catalysts also revealed that Cu/ZSM-5-15/s provides complete conversion and almost full selectivity to N$_2$ at 400–500 °C despite the shift in onset to higher temperatures. The obtained overall catalytic activity results for catalysts prepared by the ion-exchange method are in good agreement with the results for the Cu/Y and Cu ZSM-5 catalysts found in the literature [59–61]. However, the overall catalytic performance represented by both activity and selectivity is substantially improved for catalysts prepared by the sonochemical route. In the work by Peng et al., one pot synthesis of Cu/ZSM-5 with a varying Si/Al (20, 30, 40, 50, 100) and copper content ratio (1.9, 2.9, 4.5, 5.7, 8.8 wt%) was proposed. In their work, the maximal achieved conversion for 500 ppm NO$_x$ at 250 °C was equal to 98, 90 and 88% for catalysts containing 4.5, 5.7, 8.8 wt% of copper, respectively. Despite the relatively low maximum conversion temperature, it must be pointed out that at higher temperatures, the activity of catalysts decreased significantly.

The above trends, however, change when expressing activity using turnover frequency (TOF). The specific activities of the two series of USY/Y and ZSM-5-based catalysts are presented in Figure 5A,B, respectively. In the group of USY- and Y-based catalysts (Figure 5A), the Cu/Y/s catalysts prepared by sonication showed the highest activity, whereas the activity of the other catalysts decreased in the following order: Cu/Y > Cu/USY/s > Cu/USY. Among the ZSM-5 based catalysts, TOF decreased in the following order: Cu/ZSM-5-37/s > Cu/ZSM-5-37 > Cu/ZSM-5-15/s > Cu/ZSM-5-15.

*3.3. In Situ DRIFT Studies*

The comparison of the band at 1905 cm$^{-1}$ from in situ NO adsorption and dynamic TPSR NO$_x$ (Figures S10 and S11) highlights the stability of the Cu$^{2+}$-NO adduct up to 50 °C, although this band is no longer observed at temperatures above 50 °C. The detailed analysis in the 1700–1500 cm$^{-1}$ region exhibits three bands at ca. 1622, 1610 and 1575 cm$^{-1}$, which may be attributed to the nitrate or nitro species adsorbed on the catalysts surface [62]. However, due to the presence of similar vibrations in various nitrate structures, accurate band assignment is more complicated. According to Hadjiivanov [63], NO can be easily oxidized to NO$_2$ after NO and NO + O$_2$ adsorption on various oxide and zeolite catalysts. After exposing the Cu-SAPO-34 catalyst to a 500 ppm NO flow, Wang et al. [62] observed bands at 1622, 1614 and 1602 cm$^{-1}$ assigned to NO$_2$ and monodentate- and bidentate-nitrate species. Similarly, after exposing Cu/ZSM-5 to 0.45% NO/0.75%O$_2$/He, Adelman et al. [64] assigned FT-IR bands at 1628, 1594, 1572 cm$^{-1}$ to NO$_2$ coordinated to Cu$^{2+}$, and to monodentate and bidentate nitrate species, respectively. DRIFT analysis of NO adsorption on Y/USY-based catalysts showed that the band at 1622 cm$^{-1}$ is detected up to 100 °C, but disappears upon further increasing the temperature, whereas the monodentate and bidentate species remain adsorbed up to 250 °C. The results of NO$_x$ adsorption over ZSM-5-based catalysts (Figure S11) showed that small amounts of NO$_x$ are adsorbed on Cu active sites in the form of nitrates (1622 cm$^{-1}$ and 1575 cm$^{-1}$). For the Cu/ZSM-5-15 catalyst (Figure S11B), the bands at 1622 and 1575 cm$^{-1}$ are fully developed, in addition to the band at 1602 cm$^{-1}$. All bands of the Cu/ZSM-5-15 catalyst are detected because Cu/ZSM-5-15 has a 6-times higher amount of Cu than the other catalysts with the same

type of carrier (ZSM-5). Similarly, the nitrate and nitro species remain detected on the Cu/ZSM-5-15 catalyst up to 350 °C.

Exposing Cu/Y/USY and Cu/ZSM-5 catalysts to $NH_3$ (Figures S12 and S13) generates two major bands at 1617 and 1460 $cm^{-1}$, which may be attributed to ammonia adsorbed on the Lewis and Brønsted acid sites, respectively. According to the literature [62,65], Lewis acidity relates mainly to exchanged or introduced (here, Cu nanoparticles) transition metal oxides. However, extra-framework $Al^{3+}$ centers may also act as Lewis acid centers [62,66,67]. Additionally, ammonia sorption exhibited an additional band at ca. 1630 $cm^{-1}$, which may be assigned to either ammonia molecularly adsorbed on Cu or framework $Al^{3+}$ species.

Temperature-programmed DRIFT analysis of dynamic $NH_3$ sorption exhibits a continuous decrease in the relative intensities of the bands of both Brønsted and Lewis acid sites. For Y/USY-based catalysts, the band at 1630 $cm^{-1}$ is detected up to 50 °C but disappears upon further increasing the temperature, with simultaneous formation of a band at 1617 $cm^{-1}$. The band at 1630 $cm^{-1}$ may indicate weakly bound $NH_3$ on either extra-framework $Al^{3+}$ species [68] or $NH_3$ bonded to $Cu^{2+}$ sites [69].

The coexistence of the two types of Cu species on the catalysts surface was also confirmed by CO sorption experiments (Figure S8). Low-temperature desorption on weakly bonded ammonia was previously observed in $NH_3$ TPD experiments for Cu–SSZ13 [67], Cu–SAPO34 [70] and Cu–Y, Cu–USY and Cu–ZSM-5 [69]. A key piece of information is also interfered by analysis of the OH region, where ammonia desorbs at low temperatures from weak Brønsted acid sites and may be re-adsorbed on Lewis acid sites.

The results at Figure 6 shown that OH regions exhibit three main bands at 3333, 3270 and 3182 $cm^{-1}$, which may be attributed to N–H stretching vibrations of $NH_4^+$ groups (bands 3333 and 3270 $cm^{-1}$) and to ammonia adsorbed on $Cu^+$ species (3182 $cm^{-1}$) [62]. As temperature increases, these bands flatten. Similarly, the band at ca. 1630 $cm^{-1}$ assigned to weakly adsorbed $NH_3$ together with the bands at 1617 and 1460 $cm^{-1}$ decrease with the temperature. The low-temperature depletion of both bands at 1630 $cm^{-1}$ and 1460 $cm^{-1}$ is associated with desorption of weakly bound $NH_3$ rather than the re-adsorption of ammonia from weak Brønsted to Lewis acid sites.

As shown above, the dynamic adsorption of individual reactants in SCR $deNO_x$ may result in a complex DRIFT spectrum, wherein several specific catalyst properties affect the type of surface intermediates. To determine the surface intermediates during the catalytic reaction, the SCR $deNO_x$ reaction was analyzed by temperature-programed DRIFT spectroscopy (Figure S14).

The DRIFT results of the SCR $deNO_x$ reaction over USY/Y (Figure S14) do not significantly differ from the results of ammonia sorption, but only the bands at 1622 and 1460 $cm^{-1}$ are detected even though all reactants are present in the reacting mixture. Ammonia adsorbed either on Brønsted and Lewis acid sites is detected most likely due to its preferable adsorption on the surface of the catalysts, as previously reported by Knoebel and Elsner [71] at low reaction temperatures and for a $NH_3$:NO ratio of 1.1. Similar decreases in both bands at 1630 (1617) and 1460 $cm^{-1}$ are observed with the increase in reaction temperature. Additional bands originating from possible reaction intermediates have been reported in literature [62] for Cu–SAPO-34, such as $NH_4NO_2$ (band 1575 $cm^{-1}$), which may be further oxidized to $NH_4NO_3$, were not detected (band 1596 $cm^{-1}$). However, the molecular image of the catalyst surface considerably changes when exposing Cu-modified ZSM-5 catalysts to the SCR $deNO_x$ reacting mixture (Figure S15).

Under those conditions, the bands originating from Brønsted and Lewis acid sites appear. The bands at 1617 and 1460 $cm^{-1}$ are fully developed, with no band at 1630 $cm^{-1}$ possibly due to the lower Cu content of Cu/ZSM-5 catalysts. When increasing the reaction temperature up to 150 °C, new bands appear at 1596 and 1575 $cm^{-1}$, which may be associated with the presence of $NH_4NO_3$ species [62]. The formation of $NO_3^-$ and $NO^{2-}$ through $NO_2$ dimerization and disproportionation reactions leads to $NO_3^-$ and $NO^+$ species [62]. The latter is effectively replaced by $H^+$ from Brønsted acid sites and forms $HNO_3$ [72]. Indeed, ZSM-5-based catalysts exhibit an amount of Brønsted acid sites one

order of magnitude higher than Y/USY catalysts, which show the reverse trend for Lewis acid sites. However, studies have recently shown that the reaction between $NH_4NO_3$ and NO is the rate-determining step in fast SCR deNO$_x$ [62,73]. The enhanced activity of SCR deNO$_x$ proceeds according to the following reaction [73]:

$$2NH_3 + 2NO + NH_4NO_3 \rightarrow 3N_2 + 5H_2O \tag{1}$$

The onset temperature of this reaction is 170–190 °C for Fe–ZSM-5 zeolites [73]. Although we observe the $NH_4NO_3$ species on the surface of ZSM-5-based catalysts, indicating the SCR route, the activity of ZSM-5-based catalysts is shifted to higher temperatures. This may be associated with either strong interactions of adsorbed ammonia with Brønsted acid sites or a significantly lower amount of Lewis acid sites, which normally play the role of reservoir of the NO species. Conversely, the presence of the $NH_4NO_3$ on Cu-modified Y/USY catalysts cannot be ruled out. The $NH_4NO_3$ intermediate may be formed on Cu modified Y/USY, but the considerably higher amount of Lewis acid sites may provide enough NO content to reduce the $NH_4NO_3$ intermediate, as suggested by Wang et. al. [62].

### 3.4. Theoretical Results and Reaction Mechanism

SEM analysis of the Cu–Y and Cu–USY catalysts prepared by ion exchange or sonochemical irradiation showed that the zeolite surface is uniformly covered with copper in oxidized form. The UV-Vis spectra of the Cu zeolites revealed an intense and broad band at 200–400 nm and a weak and broad band at 600–850 nm with maxima at 250 nm ($Cu^+$ species), 270 nm (the $Cu^{2+}$ and oxygen atoms), 330 nm ([Cu–O–Cu]n-type clusters) and at 600–850 nm (octahedrally coordinated CuO) [48]. The theoretical models confirm the intense broad band at 200–400 nm. In the next step, oxidized Cu–O–Cu dimers form [Cu–O–Cu]n-type clusters and octahedrally coordinated CuO. NO and $NH_3$ adsorption processes were studied over Cu nanoparticles bound to zeolite clusters. Several configurations, electronic structures (charges and bond orders) and vibrations were analyzed to determine feasible pathways for the oxidation process of Cu monomers and dimers in the zeolite framework.

NO reduction cycles were analyzed as an additional step over Cu monomer and dimers under (i) water deficit conditions (Figure S10), and (ii) water-rich conditions (Figure S16(3)). The NO molecule adsorbs above terminal oxygen in Cu monomers and above bridge oxygens of $Cu_2O$ dimers. Asymmetric vibration of the NO molecule (2066 cm$^{-1}$) on CuO in Y/USY is higher than the experimental values (1947/1905 cm$^{-1}$) (Figure S16a). The vibration of NO bonded to $Cu_2O(OH)_2$ in Y/USY (1634 cm$^{-1}$) corresponds to vibrations of NO$_x$ observed during dynamic sorption experiments at low temperatures (1622 cm$^{-1}$). In ZSM-5, asymmetric vibrations of the NO molecule (1700/1598 cm$^{-1}$) over $Cu_2O$ are lower than the experimental values (1804/1622 cm$^{-1}$) (Figure S16b).

In partially hydrated or hydrated forms of Cu zeolites, the $NH_3$ molecule adsorbs on copper centers or on terminal hydroxyl groups (Brønsted acid sites), respectively. The best compatibility between theoretical models and experimental results was assessed for $NH_3$ adsorption on the $Cu_2O(OH)_2$ hydrated dimer in Y/USY: H–N–H scissoring vibrations (1616 cm$^{-1}$, Figure S17a(2)) match the main experimental values (1620 cm$^{-1}$). In H–N–H scissoring, $NH_3$ vibrations over the $Cu_2O$ dimer are slightly higher (1641 cm$^{-1}$, Figure S17a(3)) than the experimental value. Furthermore, $NH_3$ adsorption on the $Cu_2O(OH)_2$ hydrated dimer over ZSM-5 also show a high level of agreement with the experimental results (Figure S17b(2)), with N–H wagging vibrations (1437 cm$^{-1}$) matching the main experimental values (1460/1440 cm$^{-1}$).

The $Cu_2O$ dimer is essential for the formation of $N_2$. In this process, both molecules (NO and $NH_3$) adsorb simultaneously (Figure S18), in line with our SCR deNO$_x$ experiment (cf. Figures S14 and S15). The theoretical vibrations of co-adsorbed NO and $NH_3$ species match those of the low-temperature DRIFT spectra of SCR deNO$_x$ experiments for both Y/USY and ZSM-5 zeolite samples. For Y/USY zeolites, the H–N–H vibrations (1646 cm$^{-1}$, Figure S18a) show good agreement with the experimental values (1617/1630 cm$^{-1}$). For ZSM-5, the

strongest vibrations of NO with the H–N–H component at 1493 cm$^{-1}$ (Figure S12a,b) are in line with the bands observed during the experiment in the 1460–1500 cm$^{-1}$ range (Figure S15).

Our theoretical data confirm that both copper active centers and Brønsted acid sites are involved in ammonia adsorption [73,74]. Hence, two types of sites are active in the SCR process, but different mechanisms are proposed to describe their contribution in SCR reactions. Based on the concentration of Brønsted acid sites (cf. Table 1) and on recent suggestions of a unified mechanism of SCR deNO$_x$ reported by Bendrich et al. [75], we propose two different paths for (a) on the partly hydrated copper dime and (b) on the fully hydrated copper dimer. Both possible reaction paths are illustrated in Figure 8.

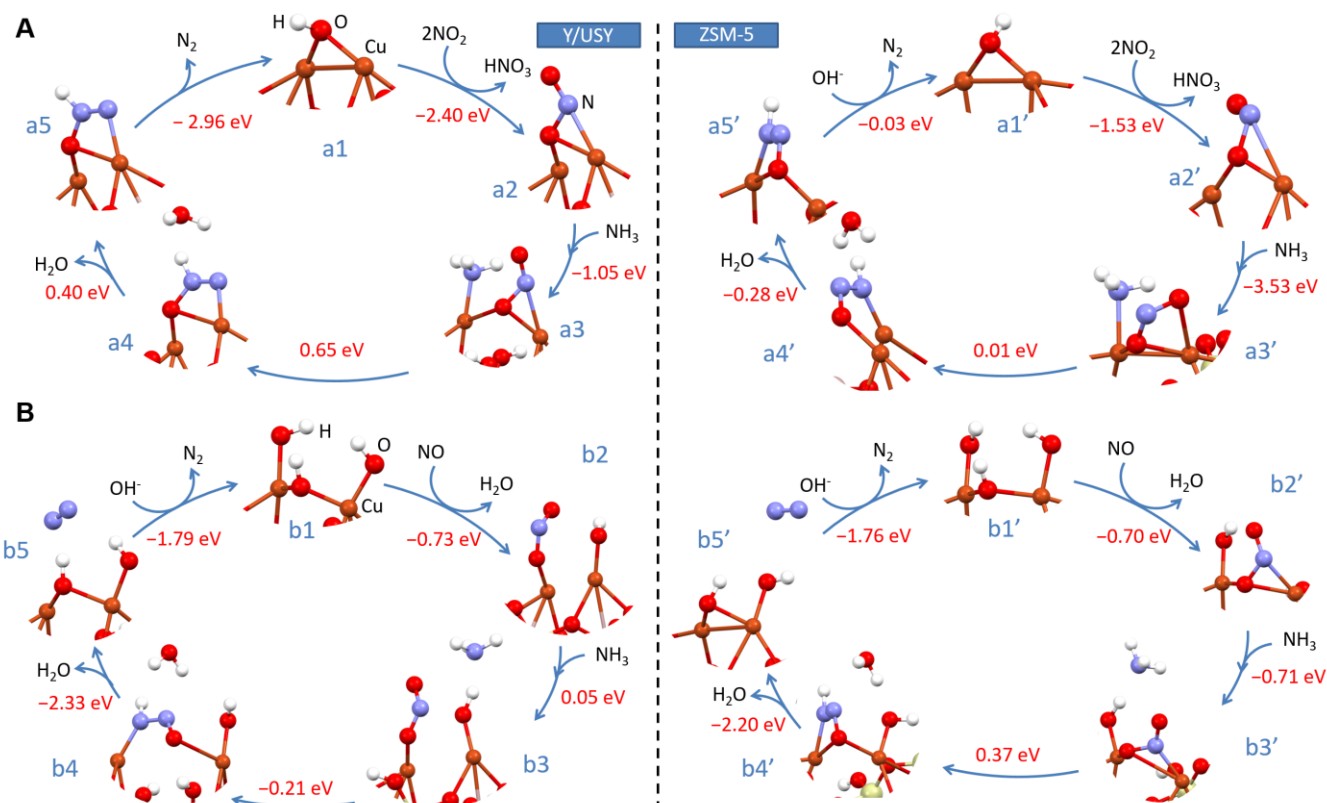

**Figure 8.** Proposed mechanism of deNO$_x$ at the Cu$_2$O dimer supported on (**A**) partly hydrated copper dimer and (**B**) the fully hydrated copper dimer. Left column: Y/USY, right column: ZSM-5.

The first path is based on the partly hydrated copper dimer (Figure 8A). The reaction starts over the partly hydrated Cu–O–Cu dimer, where only the bridge oxygen is hydrated (Figure 8(a1–a1')). In the first step, two NO$_2$ molecules react with the copper dimer, releasing nitric acid, and NO adsorbs on the bridge oxygen (exothermic, with −2.40 and −1.53 eV energy for Y/USY and ZSM-5, Figure 8(a1–a2,a1'–a2'), respectively). In the second step, ammonium is adsorbed over one of the free copper centers (exothermic reaction with energy −1.05 and −3.53 eV for Y/USY and ZSM-5, Figure 8(a3–a3')). In the next step, water and the -N–NH complex are formed in an endothermic reaction (0.65 and 0.01 eV for Y/USY and ZSM-5, Figure 8(a3–a4,a3'–a4'), respectively). The release of water is an endothermic process (0.40 eV, Figure 8(a4–a5)) for Y/USY and exothermic, with −0.28 eV energy for ZSM-5. In the last step, N$_2$ is formed, with simultaneous restoration of single hydroxyl groups on the copper dimer, which is a highly exothermic process in the case of Y/USY (−2.41 eV, Figure 8(a5–a1)) and slightly exothermic for ZSM-5 (−0.03 eV, Figure 8(a5'–a1')).

The second path occurs on the fully hydrated copper dimer (with a high concentration of Brønsted acid sites, over Y/USY and ZSM-5, Figure 8B). NO reacts with the hydrated dimer, releasing a water molecule, and the NO molecule is adsorbed on the bridge oxygen

of the Cu–O–Cu dimer (Figure 8(b1–b2) for Y/USY and b1'–b2' for ZSM-5). This process is exothermic with almost similar energy for both zeolites (−0.73 eV for Y/USY and −0.70 eV for ZSM-5). In the second step, the ammonium molecule is adsorbed on Brønsted acid sites, mainly on the hydroxyl group of the copper center (Figure 8(b2–b3)), which is a slightly endothermic process (0.05 eV) for Y/USY and an exothermic (−0.71 eV) for ZMS-5. In the next step, the hydroxyl group, $H_2O$ and the -N–NH complex are formed in an exothermic process for Y/USY (−0.21 eV, Figure 8(a4)) and surprisingly an endothermic process for ZSM-5 (0.37 eV, Figure 8(a4')). Subsequently, a -N–NH complex is generated exothermically (−1.65 eV, Figure 8(b4)) with the simultaneous release of water, which, in turn, is an endothermic process (0.62 eV, Figure 8(a4–a5)). In the last step, a $N_2$ molecule is formed, with simultaneous restoration of Brønsted acid sites on the copper dimer, which is a highly exothermic process (−3.32 eV, Figure 8(a5–a1)). Both mechanistic paths corroborate Bendrich et al. [75]. The structures b3 and b3' are similar to that observed experimentally for $NH_4NO_3$. The calculated energies for the proposed mechanisms over Y/USY- and ZSM-5-based catalysts also suggest that SCR deNO$_x$ proceeds more effectively over Cu/Y and USY catalysts prepared by sonochemical irradiation than over their Cu/ZSM-5 counterparts, thus matching our experimental observations.

## 4. Materials and Methods

### 4.1. Catalyst Preparation

The ZSM-5-type zeolite of Si/Al = 15 or 37 as well as the zeolite with Y topology of Si/Al = 4.52 (and its ultrastabilized form—USY) were prepared. For the zeolite with ZSM-5 structure, zeolite gel precursor was obtained by several steps. At the beginning, sodium aluminate (Honeywell Riedel-de Haën GmbH, Seelze, Niedersachsen, Germany, 54 wt% $Al_2O_3$, 41 wt% $Na_2O$) was dissolved in distilled water. Subsequently, tetraethylortosilicate (98%, Sigma-Aldrich, Saint Louis, MO, USA) as a silicon source and tetrapropylammonium hydroxide (1.0 M, Sigma-Aldrich) as an organic template were added to the sodium aluminate aqueous solution and then aged at room temperature for 20 h. Afterwards, the gel was transferred into Teflon-lined stainless-steel autoclaves, sealed and rotated with 56 RPM at 175 °C for 20 h. Next, the synthesized zeolites were rinsed with distilled water and dried at 80 °C overnight. In the next step, ZSM-5-type zeolite samples underwent calcination for 8 h at 480 °C with a temperature ramp of 2 °C/min in order to remove the organic template (TPAOH) from the made materials. Subsequently, thus-prepared zeolites with ZSM-5 structure type were subjected to a triple ion-exchange procedure at 80 °C for 2 h with a 0.1 M aqueous $NH_4NO_3$ solution (the reaction: $Na^+/ZSM-5+NH_4^+ \rightarrow NH_4^+/ZSM-5+Na^+$ took place). Finally, dry zeolites in ammonium form were calcined at 450 °C for 8 h in dry air, according to the following formula: $NH_4^+/ZSM-5 \rightarrow H^+/ZSM-5+NH_3$. ZSM-5-type zeolites with Si/Al = 15 and 37 in protonic form were designated as H/ZSM-5-15 and H/ZSM-5-37, respectively.

For Y-type zeolite (Si/Al = 4.52), the gel was prepared via the blending of a 2.5 M NaOH aqueous solution with sodium aluminate (Riedel-de Haën, 54 wt% $Al_2O_3$, 41 wt% $Na_2O$) and colloidal silica (Ludox AS 40%, Sigma-Aldrich, Saint Louis, MO, USA). The obtained gel was transferred into a Teflon-lined stainless-steel autoclaves, sealed and treated under hydrothermal conditions at 95 °C for 24 h. The synthesized Y-type zeolites were rinsed with distilled water and dried at 80 °C overnight.

Ultra-stabilization of Y-type zeolite (USY) was based on the triple ion exchange of Y type zeolite with a 0.1 M aqueous $NH_4NO_3$ solution at 80 °C for 2 h (according to the reaction: $Na^+/Y+NH_4^+ \rightarrow NH_4^+/Y+Na^+$), which was followed by the steaming of $NH_4^+/Y$ for 3 h at 700 °C with a temperature ramp of 2 °C/min, in the presence of saturated water vapor (1.25 kPa, 50 cm$^3$/min). The use of dry air took place during the heating and cooling steps.

For the ion-exchanged preparation method, the pre-prepared zeolite samples were placed in 0.5 M aqueous copper nitrate trihydrate solution for 24 h, and were then centrifuged and several times washed with water. The resulting powders were then dried in

a ventilated oven at 80 °C overnight and subsequently calcined at 500 °C for 4 h, using 2 °C/min.

The zeolite samples prepared by sonochemical irradiation were placed in 0.5 M aqueous copper nitrate trihydrate solutions for 20 min at room temperature. Before the sonication, the samples were outgassed for 15 min using 20 mL/min Ar flow (Linde, 99.5%) and adding 1.5 mL of ethanol to sonication solution. The sonication procedure was performed using a QSonica Q700 sonicator (Church Hill Rd, Newtown, CT, USA) with 60 W and 20 kHz power and frequency, respectively. The ultrasonic generator was equipped with a "$\frac{1}{2}$" diameter horn. The catalyst prepared sonically were subsequently dried and calcined as described above for the ion-exchange samples. In both ion-exchange and sonication preparation methods, 1.5 g of zeolite was treated with 100 mL of aqueous Cu salt solutions.

### 4.2. Catalyst Characterization

The metal content in prepared samples was determined by atomic absorption spectroscopy (Thermo Scientific ICE3000 series AAS spectrometer, Waltham, MA, USA).

The crystallinity of the prepared samples was performed by PANalytical X'Pert PRO MPD diffractometer (Philips Research, AE Eindhoven, The Netherlands) with CuKα radiation at 40 kV and 30 mA with 2θ range from 5 to 50° with a 0.033° step.

The morphology of the samples was examined on a scanning electron microscope (SEM, Nova Nano SEM 200, FEI Company, Hillsboro, Oregon, USA) equipped with an energy dispersive X-ray spectroscopy (EDX) detector for chemical analysis. The experiment was performed in low vacuum mode, using a secondary electron detector. Element distribution maps were prepared using a JEOL 5400 scanning electron microscope (JEOL Ltd., Akishima, Tokyo, Japan) with a microprobe analyzer LINK ISIS (Oxford Instrument, Goleta, CA, USA).

HRTEM analyses were performed using a JEOL NeoARM 200F atomic resolution analytical electron microscope operating at 200 kV. The microscope was equipped with a Schottky-type field emission gun, a condenser lens with Cs aberration correction and a TVIPS TemCam-F416 CMOS camera (Tietz Video and Image Processing Systems GmbH, Gauting, Germany). Alignments were performed, using the standard gold nanoparticle film method.

Porosity was determined by $N_2$ physisorption at −196 °C using a Quantachrome Nova 2000 (Quantachrome Instruments, Boynton Beach, FL, USA) surface area and pore size analyzer. The samples were outgassed in vacuum at 300 °C for 20 h before the $N_2$ adsorption measurements.

UV-Vis DR spectra were recorded on an AvaSpec-ULS3648 high-resolution spectrometer (Avantes, 7333 NS Apeldoorn, The Netherlands) equipped with a Praying Mantis High-Temperature Reaction Chamber (Harrick Scientific Co., Pleasantville, NY, USA) and a high-temperature reflection probe (FCR-7UV400-2-ME-HTX, 7400 μm fibers, Fiberdesign B.V., 7411 AH Deventer, The Netherlands), using an AvaLight-D(H)-S Deuterium-Halogen Light Source. The spectra were recorded in the 200–1000 nm frequency range. The instrument was controlled by AvaSoft v 9.0 software. The spectra were collected after dehydration at 110 °C in a helium flow of 30 mL/min.

The Raman analyses were performed by using the Witec Alpha 300 Mþ spectrometer (WITec Instruments Corp., Knoxville, TN, USA) using 600 grating and 488 nm laser line. To determine the sorption properties, the FT-IR experiments were performed using the Nicolet is10 spectrometer with an MCT detector. The scanning range of the FT-IR spectra was 650–4000 cm$^{-1}$ with 4 cm$^{-1}$ resolution. The spectra were collected by averaging 128 scans. Before the FT-IR studies, the zeolites were pressed into thin wafers and activated in vacuum at 400 °C for 1 h. CO, $NH_3$ (air products 99.5% and 99.3%, respectively) and NO (Linde 99.5%) were used as adsorbates. Small doses of adsorbates were introduced into the IR cell at room temperature (NO, CO) or at 120 °C ($NH_3$).

In situ DRIFT analysis was performed using the THERMO/Nicolet 6700 spectrometer equipped with an MCT detector and Praying Mantis High Temperature Reaction Chamber with ZnSe windows (Hidden). The spectrum of the dehydrated catalyst sample in a helium flow was used as a background. To determine the surface intermediates, dynamic DRIFT experiments were performed, as described below.

### 4.2.1. NO Evolution on Dehydrated Catalyst at Different Temperatures

The mixture of 1.0101 mol% NO/He (air products, calibration mixture, ±0.5% rel.) was fed onto the pre-oxidized catalyst surface with a total flow of 25 cm$^3$/min at 25 °C for 1 h. Then, the 11.0101 mol% NO/He mixture was switched off, and the sample was flashed with pure He, with a total flow rate of 25 cm$^3$/min and at the same temperature. After reaching stationary conditions in the reaction chamber, the spectrum was measured, and the temperature was then increased to 500 °C with a heating rate of 2 °C/min. The spectrum was recorded every 50 °C.

### 4.2.2. NH$_3$ Evolution on the Dehydrated Catalyst at Different Temperatures

The mixture of 0.9998 mol% NH$_3$/He (air products, calibration mixture, ±0.5% rel.) was fed onto the pre-oxidized catalyst surface with a total flow of 25 cm$^3$/min at 25 °C for 1 h. Then, the 0.9998 mol% NH$_3$/He mixture was switched off, and the sample was flashed with pure He, with a total flow rate of 25 cm$^3$/min, at the same temperature. After reaching stationary conditions in the reaction chamber, the spectrum was measured, and the temperature was then increased to 500 °C at a heating rate of 2 °C/min. The spectrum was recorded every 50 °C.

### 4.2.3. SCR deNO$_x$

The mixture composed of 2500 ppm NO, 2500 ppm NH$_3$ and 25,000 ppm O$_2$ diluted in pure helium (40 mL/min total flow rate) was fed to the DRIFT reaction chamber with the space time ($\tau$) of NO under these conditions, defined as $\tau = W/nNO$ (where W is the catalyst mass, and nNO is the molar flow of NO in the inlet mixture, which was 373 g h mol$^{-1}$). Before the deNO$_x$ reaction, the catalysts were preoxidized in He flow (40 cm$^3$/min) at 500 °C for 1 h. After reaching stationary conditions in the reaction chamber, the spectrum was measured, and the temperature was then increased to 500 °C with a heating rate of 2 °C/min. The spectrum was recorded every 50 °C.

Catalytic studies of selective reduction of NO with ammonia were performed in a fixed-bed quartz microreactor. The experiments were conducted at atmospheric pressure and in the temperature range of 75–550 °C. The reactant concentrations were continuously measured, using a quadrupole mass spectrometer (Tectra GmbH, Frankfurt/M, Germany) directly connected to the reactor outlet. For each experiment, 0.1 g of catalyst (particles sizes in the range of 0.160–0.315 mm) was placed on a quartz wool plug in the reactor and outgassed in a flow of pure helium at 550 °C for 1 h. Then, the gas mixture containing 2500 ppm NO, 2500 ppm NH$_3$ and 25,000 ppm O$_2$ diluted in pure helium (40 mL/min total flow rate) was used. The space time ($\tau$) of NO under these conditions was defined as above.

### 4.3. Theoretical Modeling

The theoretical modeling was performed using ab initio density functional theory (DFT) methods and a cluster model for geometry representation (StoBe) [76], RPBE functional [77,78], Gaussian function [79,80], Mulliken populations [81] and Mayer bond order [82,83]). The vibration frequencies of the adsorbed molecules were calculated by single point energy calculations of the optimized geometries. The calculations of the vibrational frequencies were performed with harmonic approximations as well as an anharmonicity fit in the Morse potential function, as implemented into the StoBe code.

The adsorption energies of the NO and NH$_3$ on the zeolite were calculated, as follows:

$$E_{ad}(NO \text{ and } NH_3/zeolite) = E_{tot}(NO \text{ and } NH_3/zeolite) - E_{tot}(zeolite) - E_{tot}(NO \text{ and } NH_3),$$

where $E_{tot}$(NO and $NH_3$/zeolite) is a total energy of the NO and $NH_3$/cluster surface complex, $E_{tot}$(zeolite) and $E_{tot}$(NO and $NH_3$) are total energies of the pure zeolite and adsorbate, respectively.

The crystalline structures of Y/USY and ZSM-5-type zeolites were downloaded from the Database of Zeolite Structure [38] and rebuilt in Powder Cell software.

The cubic phase of Y/USY is described by an $F_{d-3m}$ (# 227) space group with the following lattice parameters: a = b = c = 24.3450 Å. The unit cell consists of 706 atoms.

Zeolites with ZSM-5 structure crystallize in the orthorhombic phase and are characterized by a $P_{nma}$ (# 62) space group with the following lattice parameters: a = 20.0900, b = 19.7380 and c = 13.1420 Å. The crystal unit cell contains 201 atoms.

## 5. Conclusions

The main objective of this study was to prepare and characterize by in situ spectroscopic techniques Cu–Y, Cu–USY and Cu–ZSM-5 by ion-exchange and ultrasonic irradiation methods. The catalyst prepared by ultrasonic irradiation revealed both higher activity and selectivity in the SCR deNO$_x$ process, comparing to those prepared by the ion-exchange method. The faujasite-based catalysts prepared by the sonochemical preparation method revealed complete NO$_x$ removal in a wide 200–400 °C temperature window. On the other hand, for the copper catalysts based on ZSM-5 structure, the complete conversion was achieved in 375–500 °C. The comprehensive characterization of the prepared samples showed that the catalysts prepared by the sonochemical irradiation method are in a form of nanoparticle metal oxides, whereas for catalysts prepared by the ion-exchange method, copper remains in a cationic form. Secondly, by comparing our experimental and theoretical results, we proposed the mechanism of deNO$_x$ over Cu–Y, Cu–USY and Cu–ZSM-5 catalysts, prepared using sonochemical and ion-exchange methods. The catalytic reduction of NO was considered an ordered sequence of cycles over Cu monomer and dimers under (i) water-deficit conditions and (ii) water-rich conditions. The Cu$_2$O dimer is essential for both deNO$_x$ and N$_2$ formation, due to NO and NH$_3$ co-adsorption. The calculated vibrations of co-adsorbed species match the experimental data for both Y/USY- and ZSM-5-based catalysts. Our theoretical and experimental results highlight differences in the deNO$_x$ mechanism depending on the type of catalyst (Cu–Y/Cu–USY vs. Cu–ZSM-5), showing the following:

1.  For Y/USY-type zeolites with a relatively low concentration of Brønsted acid sites (in the 21–70 μmol/g range), those sites are essential for deNO$_x$ reaction. NO reacts with hydrated copper, thereby releasing a water molecule with simultaneous NO adsorption. Subsequently, water is desorbed, and formation of the –N–NH complex between the bridge oxygen and the copper center is observed. In the final stage, N$_2$ is formed with concurrent restoration of Brønsted acid sites of the copper dimer. In case of Y/USY, this process appears almost without barrier. For ZSM-5, the barrier is larger.

2.  For zeolites with a ZSM-5-type structure, a more energetically favorable process starts on the partly hydrated Cu–O–Cu dimer; during first step, two NO$_2$ molecules react with the copper dimer, releasing nitric acid, and NO is adsorbed on the bridge oxygen. Subsequently, ammonium adsorbs on free copper centers. Subsequently, water is desorbed, forming the –N–NH complex on the bridge oxygen. In the last step, N$_2$ is formed with concurrent restoration of the single hydroxyl group over the copper dimer. A similar path for Y/USY costs two barriers.

Taking into consideration the high amount of Brønsted acid sites (in the 173–272 μmol/g range) in the case of ZSM-5, it can be concluded from theoretical consideration that the large amount of such sites will limit the no-barrier deNO$_x$ process on the partially hydrated copper dimer in the case of ZSM-5.

**Supplementary Materials:** The following are available online at https://www.mdpi.com/article/10 .3390/catal11070824/s1, Figure S1: Cluster model of (a) Y/USY ($Al_2Si_{22}O_{66}H_{36}$) zeolite and (b) ZSM-

5 ($Al_2Si_7O_{25}H_{15}$); Figure S2: XRD of the zeolites; (A) Cu/Y and Cu/USY samples, (B) Cu/ZSM-5-15 and Cu/ZSM-5-37 samples; Figure S3: SEM images of Cu/USY and Cu/Y samples; (A) Cu/Y, (B) Cu/Y/s, (C) Cu/USY, (D) Cu/USY/s; Figure S4: SEM images of Cu/ZSM-5 samples: (A) Cu/ZSM-5-15, (B) Cu/ZSM-5-15/s, (C) Cu/ZSM-5-37, (D) Cu/ZSM-5-37/s; Figure S5: SEM mapping of Cu/USY and Cu/Y samples: (A) Cu/Y, (B) Cu/Y/s, (C) Cu/USY, D) Cu/USY/s; blue-oxygen, yellow-copper; Figure S6: SEM mapping of Cu/ZSM-5 samples; (A) Cu/ZSM-5-15, (B) Cu/ZSM-5-15/s, (C) Cu/ZSM-5-37, (D) Cu/ZSM-5-37/s; blue—oxygen, yellow—copper; Figure S7: In-situ UV-Vis spectra of prepared samples with exemplary deconvolution of Cu/Y sample (inset); Figure S8: In situ FTIR spectra of the zeolite samples with adsorbed CO at room temperature: (A) Cu/Y and Cu/USY, (B) Cu/ZSM-5-15 Cu/ZSM-5-37; Figure S9: Catalytic activity and selectivity to $N_2$: (A) Cu/Y and Cu/USY samples, (B) Cu/ZSM-5-based samples; Figure S10: In situ DRIFT spectra of $NO_x$ dynamic sorption experiments; (A) Cu/USY/s, (B) Cu/USY, (C) Cu/Y/s, (D) Cu/Y; Figure S11: In situ DRIFT spectra of $NO_x$ dynamic sorption experiments; (A) Cu/ZSM-5-15/s, (B) Cu/ZSM-5-15, (C) Cu/ZSM-5-37/s, (D) Cu/ZSM-5-37; Figure S12: In situ DRIFT spectra of $NH_3$ dynamic sorption experiments; (A) Cu/USY/s, (B) Cu/USY, (C) Cu/Y/s, (D) Cu/Y; Figure S13: In situ DRIFT spectra of $NH_3$ dynamic sorption experiments; (A) Cu/ZSM-5-15/s, (B) Cu/ZSM-5-15, (C) Cu/ZSM-5-37/s, (D) Cu/ZSM-5-37; Figure S14: In situ DRIFT spectra of SCR $deNO_x$ experiments: (A) Cu/USY/s, (B) Cu/USY, (C) Cu/Y/s, (D) Cu/Y; Figure S15: In situ DRIFT spectra of SCR $deNO_x$ experiments; (A) Cu/ZSM-5-15/s, (B) Cu/ZSM-5-15, (C) Cu/ZSM-5-37/s, (D) Cu/ZSM-5-37; Figure S16: Adsorption of NO on (a) Y/USY and (b) ZSM-5. (1) CuO monomer, (2) $Cu_2O$ dimer and (3) $Cu_2O(OH)_2$ hydrated dimer; Figure S17: Vibration of $NH_3$ adsorbed on (a) Y/USY and (b) ZSM-5. (1) $Cu_2O$ dimer and (2) $Cu_2O(OH)_2$ hydrated dimer, (3) $Cu_2O$ dimer; Figure S18: Co-adsorption of NO and $NH_3$ on $Cu_2O$ dimer over (a) Y/USY and (b) ZSM-5 zeolite.8

**Author Contributions:** Conceptualization, P.J.J. and I.C.; formulation of the scientific problem and development of experimental part, P.J.J.; development of reaction paths and geometrical models of zeolites, performing calculations and analysis of data for DFT modeling at Y/USY, I.C.; experimental tests, P.S., Ł.K.; zeolite synthesis, Ł.K.; zeolite characterization, Ł.K., L.C., R.J.J., P.J., M.S., S.G., M.M.; performing calculations and analysis of data for DFT modeling at ZSM-5, I.K. All authors have read and agreed to the published version of the manuscript.

**Funding:** Financial support was provided by the projects LIDER/204/L-6/14/NCBR/2015, 2015/17/D/ST8/01252 by NSC (PL), OP VVV "Excellent Research Teams" project no. CZ.02.1.01/0.0/0.0/15_003/0000417-CUCAM (CZ), PROM no. PPI/PRO/2019/1/00018/U/00001 (PL), PL-GRID Infrastructure.

**Data Availability Statement:** Data is contained within the article or supplementary material.

**Conflicts of Interest:** The authors declare no conflict of interest.

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
