# Peer review of "Experimental and Theoretical Studies of Sonically Prepared Cu–Y, Cu–USY and Cu–ZSM-5 Catalysts for SCR deNOx"

_catalysts, doi:10.3390/catal11070824_

Round 1
Reviewer 1 Report
This study compared the NH3-SCR catalytic activity and properties of Cu-Y, Cu-USY and Cu-ZSM-5 prepared by ion-exchange and ultrasonic irradiation method. It turned out that the sonochemically prepared catalysts possessed more nanoparticle metal oxides, and had better catalytic activity compared to the catalysts prepared by ion-exchange method. However, there are some problems in both the structure and content of this manuscript. Thus, I do not recommend to publish this article in Catalysts. The specific questions are as follows:
- The Cu content of catalysts prepared by ion-exchange and ultrasonic irradiation method is different, so the difference in catalytic activity of these catalysts might be affected by Cu content. It is better to prepared catalysts of the same Cu content if one want to discuss the influence of synthesis method.
- The content in Results section is better to be presented in Materials and Methods section, and the paragraphs depicting the content of figures in Discussion section should be in Results section.
- The main idea of this article is not clear. In the Abstract section, authors stated that the objective of this study is to prepare Cu-Y, Cu-USY and Cu-ZSM-5 by ion-exchange and ultrasonic irradiation method. In the first half of the article, the authors seems to state that ultrasonic irradiation method is better to prepare catalysts. However, in the Conclusion section, the authors summarized the reaction mechanism over different catalysts. There seems to be no main idea of the whole article.
- There are also many mistakes in format, such as the superscript and subscript.
Author Response
Dear Reviewer,
We are very appreciate for all your valuable comments and suggestions, which we tried to follow and carefully corrected our manuscript and also explain doubts in Response.
With best regards in behalf of all authors,
Izabela Czekaj

Reviewer 2 Report
This MS with the title: Experimental and theoretical studies of sonically prepared Cu-2 Y, Cu-USY and Cu-ZSM-5 catalysts for SCR deNOx was well-written and presents an interesting and insightful study.
Only some points that need revision:
The lines (118, 130, 532, 587, 589, 590) correct the English mistakes.
Figure 4 it’s not presenting Raman spectra its IR one
Figure 8 b for both, try to correct the schemes there are not clear
The zeolite samples you prepared before i suggesting you described in this MS briefly in steps
Also, the table in the supplementary file i suggesting you to insert it in the MS.
Comparing with other previous work is missing
Author Response

(The authors gave the same response as above.)

Reviewer 3 Report
This manuscript synthesized three new zeolite-based catalysts: Cu-Y, Cu-USY and Cu-ZSM-5 by using sonochemical and ion excgange methods. The structure, morphology and optical property of the obtained-samples were characterized by some characterization methods such as XRD, BET, SEM, HRTEM, Raman, UV-Vis, FT-IR and DRIFTS. Moreover, theoretical modeling methods have been performed and compared based on previously reported data of iron complexes. The improvement of catalytic reduction of NO has also been described and discussed. The experiment is systematic and the discussion is logic and convincing. Thus, I recommend the work to be published in this journal in the present form.
Author Response
Dear Reviewer,
We are appreciate Reviewer’s support of our paper and finding it clearly and transparent in the present form. We will include some modifications suggested by other Reviewers hoping it will make our paper more informative to a wide audience.
With best regards in behalf of all authors,
Izabela Czekaj

Round 2
Reviewer 1 Report
The authors properly addressed the comments and the revised manuscript has been improved. The manuscript can be accepted for the publication now.
Reviewer 2 Report
Thank you for replying